# Factors Predicting Voluntary and Involuntary Workforce Transitions at Mature Ages: Evidence from HILDA in Australia

**DOI:** 10.3390/ijerph16193769

**Published:** 2019-10-08

**Authors:** Cathy Honge Gong, Xiaojun He

**Affiliations:** 1Centre for Research on Ageing, Health and Wellbeing, ARC Centre of Excellence in Population Ageing Research, Australian National University, Canberra ACT 2601, Australia; 2College of Finance and Statistics, Hunan University, Changsha 430101, China

**Keywords:** predictors, voluntary, involuntary, workforce transitions, mature ages, Australia

## Abstract

The fast population ageing has generated and will continue to generate large social, economic and health challenges in the 21th century in Australia, and many other developed and developing countries. Population ageing is projected to lead to workforce shortages, welfare dependency, fiscal unsustainability, and a higher burden of chronic diseases on health care system. Promoting health and sustainable work capacity among mature age and older workers hence becomes the most important and critical way to address all these challenges. This paper used the pooled data from the longitudinal Household, Incomes and Labour Dynamics in Australia (HILDA) survey 2002–2011 data to investigate common and different factors predicting voluntary or involuntary workforce transitions among workers aged 45 to 64. Long term health conditions and preference to work less hours increased while having a working partner and proportion of paid years decreased both voluntary and involuntary work force transitions. Besides these four common factors, the voluntary and involuntary workforce transitions had very different underlying mechanisms. Our findings suggest that government policies aimed at promoting workforce participation at later life should be directed specifically to life-long health promotion and continuous employment as well as different factors driving voluntary and involuntary workforce transitions, such as life-long training, healthy lifestyles, work flexibility, ageing friendly workplace, and job security.

## 1. Introduction

The Australian population is ageing fast, with a predicted increase in the old age dependency ratio (the ratio of older people aged 65 years and over to the working age population aged 15–64 years) from 21 per cent in 2011 to 38–42 per cent in 2050 [1]. The rapid population ageing in Australia will lead to shortages of labour force as well as increases in government expenditure on age pensions, health and aged care services, as stated in the Australian Government’s Intergenerational Report 2010 [2,3].

Maintaining labour force participation at mature ages is considered to be the most constructive strategy for addressing all the challenges of an ageing society as working longer can not only increase productivity and tax revenue, but also assist individuals to build resources for their own retirement income as well as reduce the government’s potential liability [2,4].

However, mature age workers (aged 45–64) in Australia were found to leave employment well before pension age, and have relatively lower level of workforce participation, when compared to both domestic younger working age groups and same age groups in other countries of the Organization for Economic Co-operation and Development (OECD] [5,6]. Though the labour force participation among mature age Australians has increased substantially from 67 per cent in 2001 to 74 per cent in 2012, mainly due to the increase in females’ participation in part time jobs according to OECD Statistics [5], it was still lower than that in the United States and Canada, and much lower than that in New Zealand. Australian mature-age men saw a downward trend in participation rates that dropped from 85% in the 1960s to a low of 60% in the 1980s and 1990s, before recovering to 72% in 2011 [6]. Mature age workers were also found to be disproportionately represented among the long-term and very long term unemployed in Australia [5,7,8].

The increasing health life expectancies in last two or three decades makes labour force participation at late life more feasible, especially for those working in less physically demanding jobs. Further understanding of how ageing impacted on sustainable work ability and why workers left their paid work at later life early before the age pension age (age 65) are extremely important for workforce planning and ageing well in an ageing society.

The literature review shows that labour force participation and early retirement are complex and multidimensional [9,10,11,12,13]. Extensive attention in previous studies has been paid to individual factors from labour supply side, such as the impacts of ill health [14], financial consideration [15], joint labour supply and family care needs [16], as well as institutional factors, such as universal medical insurance, eligibility for superannuation, age pensions and income tax system [17] etc., while less studies focus on factors from labour demand side, such as employment history, work conditions, and job satisfaction [15,18,19]. Gender difference is also cognizant in literature. For instance, men are more likely to consider financial aspects, while women are more likely to consider work-life balance, such as the work and caring responsibilities and the joint retirement decision with their spouses [20].

First of all, poor health, chonic diseases, caring responsibilities, workplace inflexibility, age discrimination, without non-school qualifications and lack of trainings are found to be the major barriers to the continuous employment or reemployment of mature age workers [8,14,16]. Secondly, older people working in manual occupations are more likely to get injured or disabled, or have difficulty to meet high physical requirement when age arise, hence they are more likely to be retrenched at mature age while less likely to be reemployed in other occupations [8,21]. Thirdly, job dissatisfaction and long term unemployment are found to have strong discouraging effects on labour force participation at mature age [22,23]. Lastly, the Australian system, including the more favourable access to superannuation and age/disability pensions as well as the universal health insurance, is characterised by incentives to retire early, which might contribute to the relatively younger expected ages of retirement in Australia when compared to United States though both countries have comparable life expectancies and healthy life expectancies [17].

As there is no longer a mandatory retirement age in Australia and some other developed countries, retirement can be either voluntary or involuntary aligned with factors influencing leaving.

There are several studies in Australia looking at the individual characteristics associated with voluntary and involuntary not working [5,24]. Significant difference was found among the voluntarily and involuntarily not working groups in terms of individual/household characteristics, labour market experiences and wellbeing at mature ages [5]. In addition, ‘involuntary’ retirement is associated with a marked decline across of economic well-being measured by financial hardship, and life dissatisfaction, while there is no decline in economic welfare at anticipated early retirement [4,25,26].

Nevertheless, to our best knowledge, there is no study in Australia so far looking into the common and different factors predicting voluntary and involuntary workforce transitions at mature ages. This study aims to fill in this research gap by using the nationally representative longitudinal data drawn from the Household, Incomes and Labour Dynamics in Australia (HILDA) survey 2002–2011 for Australians aged 45–64. The results can serve as evidence to inform researchers, policy makers and industrial actions to promote workforce longevity hence to better prepare for an ageing society.

One theory and two conceptual approaches are adapted for our study to provide a useful framework to guide us to select important factors which might predict voluntary and involuntary workforce transitions at mature ages. The theory of ‘cumulative advantage’ suggests that inequalities across the life course underlie the increasing gulf between the well-off and the disadvantages in later life [27,28]. Consequently, preventing people from accumulating lifelong disadvantages in health, education and employment since their earlier life stages could help to delay the onset of chronic diseases and reduce involuntary workforce transitions at mature age [29]. The ecological model of aging [30] focuses on the “fit” between individual’s changing capacities, demands and preferences, with consequences for staying or leaving their current living environment. We modified this “fit” approach from living environment to working decision. A flexible and ageing friendly workplace could help the mature age workers to meet their changing health, financial status and preferences hence stayed at work longer. The second conceptual approach is the “elderly migration” model [31], in which “push”, “pull”, and “contextual” factors are utilized to predict older people’s decision concerning their life arrangements. The push and pull factor analysis has also been used for workforce transitions by Shultz et al. [11], in which push factors are perceived as negative considerations while pull factors as positive considerations for early retirement.

The aims of this study are to explore:

(1) How lifelong advantages and disadvantages could influence voluntary and involuntary workforce transitions at mature ages?

(2) How the changing health capacity and changing preference to work more or less hours could influence voluntary and involuntary workforce transitions at mature ages?

(3) How the pull and push factors could influence voluntary and involuntary workforce transitions at mature ages?

## 2. Materials and Methods

### 2.1. Data

This study utilizes the nationally representative Household Income and Labour Dynamics in Australia (HILDA) survey data of waves 2 to 11 (representing data for years of 2002–2011). The HILDA follows the same individuals yearly since 2011 and collect comprehensive information by asking respondents questions on socio-demographics, labour force participation, employment history, current working conditions, job satisfaction, income, housing and wellbeing, etc. [32,33].

### 2.2. Key Measures

We follow the same way in Gong and McNamara [5] and Gong and Kendig [4] to define voluntary and involuntary workforce transitions. Mature ages are defined as “aged 45 to 64 years” given that relatively few people remain in the workforce beyond age 64. Working includes both part time and full time paid work; not working includes being unemployed or not in labour force during the week before the survey. Those self-employed are excluded from this study as they have been found in literature to be very different from other workers in terms of working conditions and behaviours [34]

Voluntarily or involuntarily not working at each year was defined according to individual’s responses to four questions: (1) whether people want a job; (2) if wanting a job, whether they are looking for a job; (3) if not looking for a job, what are the main reasons; and (4) what are the main activities when they are not working.

Voluntarily not working is defined when people report that they are (1) not in the labour force, and do not want a job; (2) not in the labour force and might want a job, but they are not looking for one because of ‘does not need to work/no time/prefers to look after children/not interested’; or (3) not in the labour force, do not report whether or not they want a job, and their main activity is one of ‘retired/voluntarily inactive/study/travel/holiday/leisure/doing voluntary job’.

Involuntarily not working includes people who report that they are, (1) unemployed; (2) not in the labour force but want a job; (3) not in the labour force and want a job, but they are not looking for one because of own illness, injury or disability/childcare reason/health of someone else/too young or too old’; or (4) not in the labour force and do not report whether they want a job, and their main activity is one of ‘home duties/childcare/own illness, injury or disability/caring for ill or disabled person’.

We then define voluntary and involuntary workforce transitions as from working in one year to voluntarily or involuntarily not working in the subsequent year. We have also identified whether people had ever returned to paid work after the transitions until the last recorded HILDA wave (Wave 11 in year 2011) used for this study.

### 2.3. Methodology

The logistic and multinominal logistic models are utilized to investigate what factors jointly predicting not working, voluntary or involuntary workforce transitions among Australians aged 45–64 [35,36]. The dependent variables in the regression models are transitions from working at one year to not working, voluntarily or involuntarily not working when compared to those staying at work at the subsequent year.

Our methodology recognizes that people at mature ages could leave the workforce for a time and then returned to paid work. We count each transition from working in one year to not working in the subsequent year as an independent transition, while the change from one job to another is counted as no transition group “stayed at work”. For example, if an individual worked in one year, left work in the subsequent year, and returned to work before or until the last wave (either at a same or different job), we counted as one transition. If an individual worked in one year, left work, returned to work and left work again before or until the last wave, we counted as two transitions. The preliminary data check on HILDA survey 2002–2011 indicates that among 1241 workforce transitions in 10 years, 942 occurred on different respondents in different years with one transition per person, 260 occurred on 130 respondents with two transitions per person, and 39 happened on 13 respondents with three transitions per person. Around two thirds of these transitions were associated with not returning to paid work, while the other one third was associated with returning to paid work before last wave (year 2011).

As each individual might have zero, one, two or three transitions over ten years, there might be concern on the autocorrelation and dependence of the residuals in our regression model due to the use of pooled data of same participants from the longitudinal survey. We have implemented ways to decrease this potential bias. Firstly, every transition is treated as an independent one in our regression model with changing age, family structure, financial situation and work conditions though gender and education might be stable over time. Secondly, we use the survey data as cross-sectional one by restricting all workers aged 45–64 in each of the 10 years so that the respondents aged 45 were different in different years; Thirdly, we use the cross-sectional weights (instead of longitudinal weights) in HILDA survey which were adjusted yearly to make sure the survey data to be nationally representative in each of the years. Lastly, we have controlled for as much as the individual characteristics and taken into account the reasons for leaving paid work and main activities after workforce transitions in defining voluntary and involuntary work force transitions. Consequently, we expect the impact of the autocorrelation and dependence of the residuals on our modelling estimations to be small. Nonetheless, we estimate predictors and advise caution in any attempt to interpret our results in terms of causality [4].

As there is no large cross-sectional survey data in Australia with information on workforce transitions for older workers aged 45–64, our strategy of using pooled data drawn from different waves of existing longitudinal data could increase the number of transitions and hence be able to estimate the transition models that would not be possible by using any single year data with small sample size. This approach is not only feasible, time- and cost-efficient but also permits the study of transitions occurred in a longer period than was assessed within any single investigation. All the analyses are conducted by STATA 15.1 (StataCorp LLC, College Station, TX, USA).

### 2.4. Selected Predictors

According to the theory of ‘cumulative advantage’, ecological model of aging, and the “elderly migration” model as discussed above, we use, (1) the proportion of paid and unemployed years, tenure in current occupation, and the highest educational attainment from HILDA survey data to reflect lifelong advantages and disadvantage; (2) whether paying mortgage or whether having a child or dependent student to reflect financial status; aging and long term health conditions to reflect changing health capacity and preference to more or less work hours to reflect changing preferences; (3) long term health conditions, job dissatisfaction, fixed term and casual contracts, high local unemployment rate, working as labourers as push factors, while no mortgage, partnership, partner’s working status and income, working as professional staff or in public sector as pull factors. Part time work can be push or pull factor depending on individuals.

The multiple factors associated with work force participation at later life found in the empirical studies include physical and mental health, educational attainment, tax-transfer, expected retirement income and health insurance systems, social and employer’s attitudes to ageing, caring responsibility, work flexibility, access to retraining and support services, occupations and other job characteristics [7,12,13,14,15,16,17,22,23].

The predictors controlled for in our final regression models are: (1) individual and household characteristics, including age, age square term, gender, partnership, number of children under age 15, number of dependent students aged 15–24; (2) life-long advantages/disadvantages, including educational attainment, tenure in current occupation, ratios of paid and unemployed years after graduation; (3) financial factors, including whether paying mortgage or not, partner’s working status and income; (4) work conditions, including working in public or private sector, employment type (full time or part time), occupations (professionals, managers, technicians, administration staff, operators/drivers/labourers etc.), contract type (permanent, casual or fixed term); (5) Changing capacity and preference, including long term health conditions, preference of work hours (same, more or less hours); (6) job dissatisfaction on various job aspects (job payment, job security, work itself and working hours); and (7) state average unemployment rate at the same year when workforce transitions occurred.

Job satisfaction/dissatisfaction is measured by a group of questions in HILDA asking individuals “How satisfied are you with your job (overall, job payment, job security, work itself and working hours) on a scale of 0 (the most dissatisfied) to 10 (the most satisfied)?” In this study, we used the job dissatisfaction with four job aspects (instead of the satisfaction level on job overall). We generate a dummy variable each aspect of job satisfaction: dissatisfied if with a response from 0 to 5 and satisfied if with a response from 6 to 10. The scale (6 out of 10) is used as a threshold for being satisfied or not in this study as 60 out of score 100 is socially perceived as a threshold of being satisfied or not during school evaluation. In addition, the distribution of job satisfaction (as shown in the Appendix A) shows that the proportion of respondents with a satisfaction level lower than score 6 was ranging from 10% to 20% in HILDA 2011, which is more close to the proportion of leaving paid work (about 6%) when compared to the proportion with a satisfaction level lower than the mean or the median (30–50%).

Long term health condition is measured by asking respondents “Does anyone here have any long-term health condition, disability or impairment, as shown in the showcard (HF7) (1 = yes, 2 = no”? These include sight problems not corrected by glasses/lenses, hearing problems, speech problems, blackouts, fits or loss of consciousness, difficulty learning or understanding things, limited use of arms or fingers, difficulty gripping things, limited use of feet or legs, a nervous or emotional condition which requires treatment, any condition that restricts physical activity or physical work (e.g., back problems, migraines), any disfigurement or deformity, any mental illness which requires help or supervision, shortness of breath or difficulty breathing, chronic or recurring pain, long term effects as a result of a head injury, stroke or other brain damage, a long-term condition or ailment which is still restrictive even though it is being treated, any other long-term condition such as arthritis, asthma, heart disease, Alzheimer’s, dementia, etc.

For comparison purpose, we have adjusted income data into 2009 price by indexing them using the average of monthly Australia national Consumer Price Index (CPI) for each year. The annual unemployment rate by state WAS generated by the average of ABS monthly unemployment rates [37]. We use whether people are still paying their mortgage as a proxy for their financial status, as wealth data is only available in every four years in HILDA.

## 3. Results

### 3.1. Incidence of Workforce Transitions

As shown in the first part of Table 1, the ten-year pooled data yields approximately 1241 work force transitions from working at one year to not working in the subsequent year: voluntarily not working (626, or 3.43%), or involuntarily not working (556 or 3.05%), when compared to staying at work (17,064, or 93.5%). As shown in the second part of Table 2, there are: (1) 490 voluntary transitions without returning back to work till last wave (year 2011); (2) 136 voluntary transitions returning back to work until last wave; (3) 282 involuntary transitions without returning back to work till last wave; and (4) 274 involuntary transitions returning back to work until last wave. After voluntary not working, only one fifth returned back to work, while after involuntarily not working, around half returned back to work. This is mainly due to financial pressure after leaving their jobs [4,38].

Figure 1 provides the age profiles of voluntary and involuntary workforce transitions from one year to the subsequent year occurred within the survey period of 2002–2011. It shows that at age 45, people start to exit their jobs slowly and gradually, either voluntarily or involuntarily, with a relatively higher proportion of involuntary workforce transitions than that of voluntary workforce transitions. Since age 53, the voluntary workforce transitions increase much more rapidly while the proportion of involuntary workforce transitions are relatively constant; and voluntary workforce transitions starts to overwhelm involuntary workforce transitions. For instance, the proportions of voluntary and involuntary workforce transitions are 1.27 per cent and 2.31 per cent at age 45 years, 3.13 per cent and 2.71 per cent at age 53 years, and 18.84 per cent and 5.31 per cent at age 64 years.

### 3.2. Individual Characteristics and Work Conditions

Table 2 presents the individual characteristics and work conditions associated with workforce transitions which are used in the final regression model. It shows that the total number of workforce transitions from working to not working is 16,811, in which, 15,701 were staying at work, 563 voluntarily not working, 492 involuntarily not working, 55 were not working but ‘unable to be determined’ as voluntary or involuntary one.

When transitions occurred, the average age of workers in our study was 52.23 years. About 51 per cent of them were males, 76 per cent currently had a partner while 24 per cent did not have a partner (never married, or previously with a partner). On average, there were 0.31 children (younger than 15 years) and 0.34 dependent students (aged 15–24 and at school) per household. About 38 per cent of mature age workers had a degree/diploma, 25 per cent with a certificate, 10 per cent with year 12 completion and 27 per cent finishing year 11 or below. The proportion with any long-term health condition was about 21 per cent.

Regarding working or not at mature ages, the most important financial concerns are whether still paying mortgage, eligibility to superannuation, whether partner is working or not and by how much income [39,40]. In our study, 44 per cent of mature age workers were still paying their mortgages, 32 percent of them were eligible for superannuation and only 0.31 percent of mature age workers did not have any superannuation (as age and age square term are highly correlated to whether eligible for superannuation, hence we have removed the variable “whether eligible for superannuation” from the final regression). Among those with a partner, 80 per cent of their partners were working, and the average annual income of working partners was AU$74, 230 at 2009 price.

When transitions occurred, the average tenure in current occupation was 14.37 years. After graduation, on average, 87 per cent of years after graduation were paid years and 2 per cent were unemployed years. 35 percent of respondents were working in public sector and more than two thirds (73 per cent) were working full time and 27 per cent working part time. About 38 per cent were managers and professionals, 11 per cent were technicians; 34 per cent were workers, sales, clericals or administrative staff; and 17 per cent were operators/drivers/labourers. The majority (76 per cent) had a permanent or ongoing contract, 9 per cent had fixed term contract, 15 per cent were working on a casual base, and very few were on other contract types. More than half of people (58 per cent) preferred to work the same hours as they currently did, about one third (32 per cent) would like to work less hours, and only one tenth (11 per cent) preferred to work more hours. About one fifth of workers were dissatisfied with their jobs, in which 19 per cent, 13 per cent, 11 per cent and 18 per cent were dissatisfied with their job payment, job security, work itself and working hours, respectively. The state average unemployment rate is 5.2 per cent across all years.

### 3.3. Regression Results

We have run three multivariate regression models: Model 1 is the logistic model on the transitions from working to not working; Model 2 is the multinominal logistic model on the transitions from working to voluntarily or involuntarily not working; and Model 3 on the transitions from working to (1) voluntarily not working till last wave, (2) voluntarily not working and back to work, (3) involuntarily not working till last wave, or (4) involuntarily not working and back to work. The three models all used workers staying at work from one year to the subsequent year as their reference group.

The estimated coefficients and significant levels of all predictors from the three models are reported in Table 3 for comparison (the full models with all the estimated coefficients, standard deviations and significance levels are reported in Appendix A. Marginal effects were also calculated but not reported here, and are available on request to the corresponding author Dr Cathy Gong.) The first column of numbers reports estimated coefficients from Model 1, the second and fifth columns present estimates from Model 2, and other columns from Model 3.

Table 3 shows that: (1) age squared term significantly predicts voluntarily not working and not going back to work, while it is associated with less likelihood to voluntarily leave paid work and going back to work. (2) Age is insignificant to involuntarily not working (no matter going back to work or not). (3) Males are more likely to voluntarily not working and going back to work. (4) Currently without a partner decreased voluntarily not working (no matter going back to work or not), but it increased involuntarily not working and going back to work. (5) The number of dependent students decreased voluntarily not working and not going back to work, as well as decreased involuntarily not working and going back to work. Education is insignificant to all the work force transitions (no matter they are voluntary or involuntary, and no matter going back to work or not).

Long-term health conditions significantly increased both voluntarily and involuntarily not working (no matter going back to work or not). Besides health, finance is also a very important factor in explaining work force transitions at later life. We found that, (1) still paying mortgage decreased voluntarily not working (no matter going back or not), but it was insignificant to involuntary not working. (2) Having a partner who is working decreased voluntarily not working (no matter going back to work or not), and it decreased involuntarily not working and not going back to work. (3) Partner’s income only slightly increased voluntary not working (no matter going back to work or not), and also slightly increased involuntarily leaving paid work and going back to work.

Both work conditions and job dissatisfaction predicted voluntary and involuntary not working at later life, but in different ways: (1) Tenure, as defined as years in current occupation, increased voluntary not working, while decreased involuntary not working. (2) Proportion of paid years decreased both voluntary and involuntary not working, while proportion of unemployed years significantly increased involuntarily not working (no matter going back to work or not). (3) Working in public sector significantly decreased involuntary not working (no matter going back to work or not). (4) Working part time increased voluntary not working (no matter going back to work or not), as well as increased involuntarily not working and not going back to work). (5) Workers/sales/clericals/administrative staff/drivers/labours were more likely to voluntarily not working and not going back to work, while less likely to voluntarily leave paid work and going back to work. (6) Fixed term contract significantly predicts involuntary not working (no matter going back to work or not), while casual work predicts both voluntary and involuntary not working (no matter going back to work or not). (7) Preference to work less hours significantly predicts voluntary not working (no matter going back to work or not), and increased involuntary not working and going back to work, while preference to work more hours predicts less voluntary not working (no matter going back to work or not) while increased involuntarily not working and going back to work. (8) Dissatisfaction on job security and work itself predicts involuntary not working (no matter going back to work or not); while dissatisfaction on work hours predicts voluntary not working and not going back to work.

In order to better understand how different factors could drive voluntary and involuntary work exits at later life, we compared the signs of estimated coefficients of predictors from the regression Model 2 (Table 4). It demonstrates that the factors driving voluntary and involuntary workforce transitions at mature ages are very different in Australia excepting that both long term health conditions and preference to work less hours increased while having a working partner and proportion of paid years decreased both voluntary and involuntary work force transitions. Besides these four common factors, the voluntary workforce transitions were jointly driven by individual and household characteristics, financial concern, employment history and current work conditions; while involuntary workforce transitions were mainly driven by vulnerable employment history and current work conditions (Table 4).

## 4. Discussion

### 4.1. Summary of Findings

Findings in our study indicate that the working proportion decreases slightly since age 45 and then goes down rapidly after age 50, especially after age 55. The majority of mature age workers moved from working to voluntarily not working and only a few to involuntarily not working. Before age 53, the proportion of involuntary workforce transitions is slightly higher than that of voluntary workforce transitions. While after age 53, voluntary workforce transitions increase rapidly and start to overwhelm involuntary workforce transitions. The positive age effect on voluntary workforce transitions, especially after age 53 years, is likely to be associated with the rising opportunities to be eligible to use income from superannuation or receive a disability pension [41]. Once other factors have been controlled for, the age effect is insignificant for involuntary workforce transitions, indicating that the observed slight increase in involuntary workforce transitions with age in Figure 1 is unlikely to relate to age itself.

It is found that there are four common factors (long term health conditions, prefer to work less hours, having a partner working and proportion of paid years) which had significant impacts on both voluntary and involuntary work force transitions. In which, proportion of paid years played the most important role, followed by long term health conditions, having a partner who is working, and prefer to work less.

Both historical and current employment statuses have significant impacts on workforce transitions and this is consistent to the accumulated life-long advantage and disadvantage theory used for labour market [27,42]. The longer the tenure in current occupations, the less likelihood to exit paid work involuntarily while the higher probability to exit paid work voluntarily. The higher the proportion of paid years after graduation, the lower likelihood of not working either voluntarily or involuntarily at later life, while the higher the proportion of unemployed years, the higher probability of involuntary not working at later life.

Having long term health conditions has a similar and strong power in predicting both voluntary (not going back to work), and involuntary workforce transitions (no matter going back to work or not). This reflects the fact that long term health condition is a major reason for mature age workers to leave paid work, as well as a barrier for them to go back to work. This could be explained that mature age workers with long term health conditions might value their free time more and are more likely to be eligible for government disability pension hence have a higher probability to leave their paid work voluntarily. On the other hand, workers with long term health conditions are less demanded by their employers hence their probability of involuntarily not working is higher.

Family structure, partnership and paying mortgage also have significant impacts on workforce transitions at mature ages. Unsurprisingly, home buyers still paying mortgage are less likely to exit paid work voluntarily. Workers, currently without a partner, are less likely to exit paid work voluntarily and are more likely to go back to work after involuntary workforce transitions, reflecting their high independence in both time and finance. Workers with a partner who is working are less likely to voluntarily exit paid work, reflecting the complementarities of joint arrangement of work and leisure time between partners at later life. Partner’s income had a significant, positive but small impact on voluntary workforce transitions. Workers with dependent students are less likely to exit paid work voluntarily, but once they exit their paid work involuntarily, they are less likely to come back to work, reflecting that they have strong incentives to stay at work to support their dependent students but face strong barriers to go back to work at mature ages.

There were also strong incentives among mature age workers to adjust their working hours when age arises, and there is room to improve workforce participation by hours for those who were under employed. When compared to those who prefer to work same hours, prefer to work less hours significantly predicts both voluntary and involuntary not working; dissatisfaction on working hours increased voluntary not working, while prefer to work more decreased voluntary not working. Part time, casual work, and prefer to work less hours might be a signal for a pathway to retirement for those workers who were financially prepared [43]. While for workers with an overwhelming workload, work hours need to be adjusted down according to their changing health, capacity and preference.

The estimated coefficients of other predictors are mostly under our expectations and in line with existing literature. For instance, workers who were dissatisfied with their job security or work itself, working under fixed term or casual contracts were more likely to leave paid work involuntarily while working in public sectors predicts a lower probability of involuntary workforce transitions. Non-professional staff (workers, sales/clericals, administrative staff, operators, divers, labours) had a lower likelihood to go back to work after voluntary not working.

The general insignificance of education levels on workforce transitions at mature ages might simply reflect the combination of income and substitution effects as mentioned by human capital theory. On the one hand, the income effect predicts that higher educated people with higher earnings are more likely to be able to afford to enjoy free time by exiting their paid work earlier before the age pension age. On the other hand, the substitution effect states that higher educated people might stay in paid work longer due to a higher opportunity cost driven by their higher wages or better employment conditions [8,44].

### 4.2. Policy Implications

In order to remove the barriers that many older workers are facing to carry on working, the OCED called on the Australian authorities to take further actions to enhance the public awareness and effectiveness of age discrimination legislation, to prevent social securities as incentives to early retirement and to strengthen older workers’ employability [6]. It is found that in New Zealand from 1992 to 2001, increasing age pensions age from 60 to 65 for both men and women, and allowing people to stay at paid work while receiving age pensions have effectively increased labour force participation at mature and older ages [6].

The OECD has concerns that in Australia, the possibilities to draw superannuation benefits unconditionally as a lump-sum at an early age, to use disability pension as a pathway to early retirement, and to reduce income from age pensions while receiving income from paid work might contribute to the decrease of labour force participation at mature ages, which have not been well examined.

The existing policies and current efforts to increase workforce participation at later life in Australia include improving education and training, assisting attachment to labour market, enhancing long term campaign with age stereotypes and age discrimination, financially encouraging employers to hire workers aged 50 and plus, providing substantial superannuation tax incentives after age 60, as well as to increase progressively the preservation age for superannuation from 55 to 60 and age pension eligible age from 65 now to 67 and further to 70 [2,21,45]. The current policies aiming to increase age eligibility for superannuation and age pensions are expected to delay some voluntarily not working with financial consideration but not for others without financial consideration.

The implications of our findings suggest that in order to facilitate longer workforce participation and enhance productivity and well-being at later life, different government policies and employment practices should be engaged to address the major causes of voluntary and involuntary workforce transitions at mature ages. Promoting mature age workers’ health, employability, work flexibility and friendly work environment, as well as providing rational and secure pathways before full retirement could help older workers to meet their changing health and family needs, hence stay in paid work longer [4,46,47,48].

A central and new challenge for an ageing society is to enable continued workforce participation at later life by preventing or ameliorating chronic diseases or disabilities when age arises for workers in their 50s or 60s [49]. The ‘health first’ and “fitness” approaches, and life-long accumulative disadvantage theory, should be taken in consideration in making further policies to tackle the health-related or vulnerable worklessness for mature aged workers in Australia.

For voluntarily not working at later life, the ‘health first’ and “fitness” approaches suggest that the fundamental policies and employment practices are health promotion over life span, age-friendly workplaces, work flexibility to meet changing preference etc. The “health first” approach targets the root cause of worklessness through preventing, improving and managing chronic diseases at mature ages hence refining workers’ health capacity and employability [50]. Health promotion can reduce both local worklessness and health inequalities but need joint efforts among clinical groups, work program providers work organizations and local authorities [4,51].

In Europe where some of countries have already taken important steps to tackle the challenge of ageing population, health promotion activities in workplace for ageing workers have recently been promoted as a new approach to improve occupational and populational health [51]. Besides the historic approach that takes into account occupational risks, technical and medical expertise, and ergonomic adaptions in the work environment, this new approach promotes healthy habits which may delay the onset of diseases or help to manage the chronic diseases. A great effort has been made to increase the motivation of older workers to move to healthier habits, as older workers are more reluctant to make changes than younger workers. The existing health interventions in Australia have mainly occurred in communities for promoting physical activities and in primary care centers for disease prevention, as well as in health and aged care institutions for managing disabilities and chronic diseases, the health promotion activities in workplace are still inadequate. Australia shares the same obstacle for health promotion programs as in Europe on how to ensure continuity of funding and effectiveness after the end of the intervention programs.

With the extended life expectancy, many people in their 50s and 60s are expected to work with some form of mild chronic diseases or disabilities. It has been found that working longer hours than what is feasible is harmful to health, indicating that the standard full time work might no longer be the best fit for many older people with long term health conditions [49,52,53]. The “fitness” approach encourages the employers and employees to discuss and negotiate work hours, hourly wage, and other work arrangement after reaching certain ages according to the changes in both labour supply and demand sides hence to achieve a new agreement which could be beneficial to both employers and employees. Our analysis supports this by evidence that almost one third of mature age workers had indicated their willingness to reduce working hours at later life. The alternative options could be secured part time or casual employment as a pathway to full retirement when health and energy declined at mature ages.

For involuntarily not working at later life, the life course approach suggests the implementation of policies in equalizing lifelong opportunities for health, education, training, retraining, employment assistance, can help to promote lifelong health, work capacities and continuous labour force participation for those with vulnerabilities, such as those without non-school qualifications or trainings.

Job insecurity is the most emerging issue to be addressed for those involuntarily not working. In last two or three decades there has been continuous shift away from secure (permanent) to insecure (fixed term or casual) jobs and from standard full time contracts to non-standard arrangements (part time, casual, short term or irregular hours) in Australia and many other developed countries [29,49] These changes are mainly driven by the economic restructure and technical change resulting a shift of economy from manufacturing sectors to service sectors [29,49]. It has been found that during this shift, older workers are often forced into precarious employment with the consequent cycle of fewer job opportunities, little training and lack of income security, exposure to discrimination, harassment and workplace bullying, non-portability of leave entitlements, as well as a reduced capacity to exercise autonomy in how the work is done, resulting damage to health and wellbeing [29,49,54]

The new policy making will also need to take into account the changes in role model and work culture during the new economy, as these changes are likely to increase the demand for secure part time jobs at mature ages. On the labour force supply side, our analysis shows that the current generation of mature age workers are baby boomers who are still working under the male breadwinner female caregiver model, pursuing the norm of “good jobs” as secure and full time jobs, working as long as possible except for married females who participated in labour force flexibly to balance work and care needs [29,49,55,56,57]. While for the new generations, the working model is changing significantly where people spend more years on formal education and trainings in early life in order to better survive in the new knowledge-based economy, with males working mostly full time but spending more time with children and reducing work effort in response to health change in later life, while females reducing working hours instead of fully withdrawing from work force for caring responsibilities [49]. On the labour force demand side, the employer and recruiter perspectives are also changing in terms of what they want from mature age workers. For instance, the notion of employability has changed from “reliability, punctuality and the ability to accept direction” to “resourcefulness, adaptability and flexibility”, the requirement for formal qualification has grown, the low-skilled jobs have been reduced and casualization of the workforce is increasing in an uncertain economy [29].

Current efforts in Australia, to increase job security of part time or casual work, include extra loading in earnings, wage subsidy, and protection against unfair dismissal etc. [49]. Further investigation needs to be done on how to increase the options of more secure part-time, fix-termed or casual employment and other forms of more tenuous engagement in a new economy of services. Retraining programs with focus on broader skills and employability for workers who have lower level of qualifications and fewer training opportunities in a declining industry or occupation could assist transition from one industry to another hence to ensure lifelong continuous labour force participation [8].

As discussed, the strategy of using pooled data drawn from different waves of existing longitudinal data allows us to estimate the workforce transitions at mature ages that would not be possible by using any single year data with small sample size. Nonetheless, we estimate predictors and advise caution in any attempt to interpret our results in terms of causality [4]. Other limitations of this study include the utilization of one-digit occupational group to identify whether the work is physically demanding, and the use of job dissatisfaction to identify potential psychosocial risk factors in the workplace. For future research, more detailed occupational categories in HILDA can be used to identify possible impacts of hazardous work and impairing work on work force transitions. In addition, work-related stress [58] and workplace psychological harassment [59] are recognized world-wide as major challenges to workers’ mental health problems and other stress-related disorders, which are known to be among the leading causes of early retirement from work, high absence rates, overall health impairment [58,59]. Larger cross-sectional and longitudinal survey for mature age and older workers are necessary for further estimations.

## 5. Conclusions

Encouraging mature age workers to work as long as possible is a long-term strategy in Australia and many other countries to address the policy challenges associated with ageing population. The issue in the current labour market is that mature age workers are facing long-term unemployment or underemployment while employers increasingly claim labour force shortages [29]. This calls for a deep understanding of what mature age workers need and face, what policies can address informed by a life-course perspective and a move away from the focus on individuals to the attitudes of society and employers to older people, including job security, promotion of healthy life styles, ageing friendly workplace and work flexibility [8,29,49,51].

Our research indicates that many mature age Australians want to work longer, and continuous paid employment could help older workers to work longer at later life, but how to best facilitate the retirement transitions when health deteriorates and how to keep older workers to work decently, safely and appropriately will be the challenges for government, industry and society as a whole to overcome.

## Figures and Tables

**Figure 1 ijerph-16-03769-f001:**
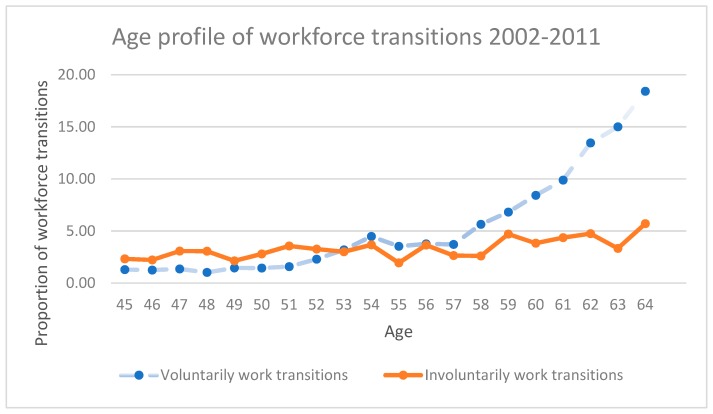
Age profile of voluntary and involuntary workforce transitions, Australia 2002–2011. Data source: Authors’ own calculation from the pooled data of HILDA survey 2002–2011.

**Table ijerph-16-03769-t001a:** **Panel A.**

The Base Year	The Subsequent Year	Number of Workers at the Base Year	Same Participants at the Subsequent Year	Staying at Paid Work	Left Paid Work	Voluntarily Left Paid Work	Involuntarily Left Paid Work	Left Paid Work But Can Not Be Identified
2002	2003	1845	1741	1618	123	58	54	11
2003	2004	1929	1810	1659	151	68	62	21
2004	2005	1906	1818	1689	129	62	65	2
2005	2006	2021	1945	1817	128	76	50	2
2006	2007	2132	2050	1919	131	68	59	4
2007	2008	2214	2141	2030	111	60	46	5
2008	2009	2291	2211	2061	150	67	76	7
2009	2010	2366	2281	2119	162	87	74	1
2010	2011	2399	2308	2152	156	80	70	6
Sum of transitions				17,064	1241	626	556	59

**Table ijerph-16-03769-t001b:** **Panel B.**

The Base Year	The Subsequent Year	Voluntarily Left Paid Work Until Last Wave	Voluntarily Left Paid Work but Back Before Last Wave	Involuntarily Left Paid Work Until Last Wave	Involuntarily Left Paid Work but Back Before Last Wave
2002	2003	42	16	13	41
2003	2004	53	15	29	33
2004	2005	42	20	22	43
2005	2006	51	25	23	27
2006	2007	51	17	27	32
2007	2008	44	16	25	21
2008	2009	56	11	36	40
2009	2010	71	16	37	37
2010	2011	80	0	70	0
Sum of transitions		490	136	282	274

Source: Authors’ own calculation from HILDA survey 2002–2011. Notes: (1) The first part of this table (panel A) presents the number of work force transitions by voluntarily or involuntarily not working occurred in every year and the sum of these transitions when compared to staying at paid work; (2) the second part of this table (panel B) reports the number of voluntary or involuntary work force transitions by whether returning back to paid work until the last HILDA wave (year 2011) and the sum of the transitions; (3) As shown in the first part of this table, (panel A), there were about 1182 (626 + 556) total voluntary and involuntary workforce transitions occurred in 10 years. The sum of transitions is the sum of the lines in every wave in the same part of this table; (4) the number of work force transitions is different from the number of participants. There were 106–161 workers with voluntary or involuntary transitions from working to not working within subsequent year but they were occurred on different people. (5) One participant might have zero, one, two or three transitions within 10 years. Among 1241 workforce transitions in 10 years, 942 occurred on different respondents in different years with one transition per person, 260 occurred on 130 respondents with 2 transitions per person, and 39 happened on 13 respondents with 3 transitions per person.

**Table 2 ijerph-16-03769-t002:** Individual characteristics and work conditions associated with workforce transitions.

Variables	Number of Workforce Transitions	Mean or Proportion (Weighted)
All workforce transitions	16,811	
Workforce transitions (defined)	16,756	
(1) Staying at work	15,701	94%
(2) Voluntarily not working	563	3%
(3) Involuntarily not working	492	3%
‘Unable to determine’ not working group	55	
**Individual characteristics**		
Age	16,811	52.23
Male	8574	51%
Currently without a partner	4035	24%
Currently with a partner	12,776	76%
Number of children (<age 15)	16,811	0.31
Number of dependent students (aged 15–24)	16,811	0.34
**Educational attainment**		
(1) Degree/diploma	6690	38%
(2) Certificates	4156	25%
(3) Year 12 or equivalent	1509	10%
(4) Year 11 or below	4456	27%
With long term health condition	3530	21%
**Financial status**		
Paying off mortgage	7397	44%
Eligible for superannuation	5380	32%
No super	52	0.31%
With a working partner	12,452	80%
Partner’s income ($1000)	12,452	74.23
**Work conditions**		
Tenure (years)	16,811	14.37
Proportion of years with payment	16,811	87%
Proportion of years unemployed	16,811	2%
Public sector	5884	35%
**Employment type**		
(1) Full time employee	11,931	73%
(2) Part time employee	4880	27%
**Occupations**		
(1) Manager/professional	6585	38%
(2) Technician	1718	11%
(3) Worker/sales/clerical/admin	5879	34%
(4) Driver/labourer	2629	17%
**Contract type**		
(1) Permanent	12,550	76%
(2) Fixed term	1551	9%
(3) Casual	2651	15%
**Preference**		
(1) Prefer less work hours	5459	32%
(2) Prefer same work hours	9521	57%
(3) Prefer more work hours	1831	11%
**Job dissatisfaction with job aspects**		
(1) Unsatisfied: job payment	3194	19%
(2) Unsatisfied: job security	2185	13%
(3) Unsatisfied: work itself	1849	11%
(4) Unsatisfied: working hours	3026	18%
State average unemployment rate	16,811	5.2%

Source: Authors’ own calculation using the pooled data of HILDA survey 2002–2011. Notes: (1) The number of workforce transitions in this table (16,811) is from the final regression model hence is slightly less than the total number of transitions presented in Table 1 (17,064), due to missing information for some predicting variables. (2) The eligible age for superannuation in Australia ranges from 55 to 60 based on individual birth cohorts: age 55 if born before 1960; age 56, 57, 58 and 59 if born in 1960–1963; age 60 if born after 1963. (3) We use whether paying mortgage to represent financial status as we found that there is no significant difference among outright owners and renters regarding their probability of workforce transitions. The owners have home ownership and relatively higher wealth but the renters are more likely to receive government rent allowance once they are not working.

**Table 3 ijerph-16-03769-t003:** Estimated coefficients and significance levels for all predictors of workforce transitions.

Years 2002–2011	Working to not Working	Voluntarily Work Exits	Voluntarily Work Exits till Last Wave	Voluntarily Work Exits and Back to Work	Involuntarily Work Exits	Involuntarily Work Exits till Last Wave	Involuntarily Work Exits and Back to Work
	Model 1	Model 2	Model 3	Model 3	Model 2	Model 3	Model 3
**Individual characteristics**							
Age-45	0.018	0.04	0.066	0.117	0.042	0.094	0.04
(Age-45) squared term	0.004 ***	0.004 **	0.005 *	−0.007 *	0.001	0	−0.001
Male	−0.031	0.056	−0.138	0.91 ***	−0.09	0.14	−0.332
Currently without a partner	−0.305 **	−0.673 ***	−0.668 ***	−0.865 ***	0.166	−0.182	0.601 **
Number of children	−0.063	−0.097	−0.108	−0.127	−0.054	−0.188	0.064
Number of dependent students	−0.247 ***	−0.275 *	−0.342 *	−0.198	−0.165	0.05	−0.363 **
**Education**							
(1) Uni degree/diploma							
(2) Certificates	0.05	0.117	0.085	0.204	0.131	0.004	0.26
(3) Year 12 or equivalent	0.161	0.334	0.285	0.448	−0.013	−0.045	0.026
(4) Year 11 or below	−0.097	0.035	0.022	0.087	-0.083	0	−0.165
With long-term health conditions	0.648 ***	0.657 ***	0.753 ***	0.286	0.624 ***	0.642 ***	0.616 ***
**Financial factors**							
With mortgage	−0.158 *	−0.392 **	−0.42 ***	−0.287	0.061	0.111	0.018
Partner is working	−0.546 ***	−0.652 ***	−0.576 ***	−0.959 ***	−0.291 *	−0.46 **	0.023
Partner’s income	0.002 ***	0.002 ***	0.002 ***	0.002 **	0.001	−0.001	0.002 ***
**Job conditions**							
Tenure in current occupation	0.001	0.009 *	0.008	0.013	−0.012 *	−0.009	−0.015
Proportion of paid years	−1.192 ***	−1.662 ***	−1.5 ***	−2.181 ***	−0.744 **	−0.869 *	−0.664
Proportion of unemployed years	1.446 ***	−0.227	0.08	−1.145	2.318 ***	2.88 ***	1.828 **
Public sector	−0.277 ***	−0.075	−0.065	−0.097	−0.695 ***	−0.51 **	−0.912 ***
**Employment type**							
(1) Working full time							
(2) Working part time	0.585 ***	0.909 *	0.762 ***	1.597 ***	0.223	0.563 ***	−0.125
**Occupations**							
(1) Managers/professionals							
(2) Technician	0.25	0.172	0.328	−0.379	0.303	0.378	0.24
(3) Workers/sales/clericals/admin. Staff	0.046	0.113	0.296 *	−0.497 **	0.005	-0.022	0.037
(4) Drivers/labourers	0.152	0.201	0.501 **	−1.027 ***	0.11	0.059	0.178
**Contract type**							
(1) Permanent/ongoing							
(2) Fixed-term	0.489 ***	0.17	0.169	0.242	0.841 ***	0.712 **	0.943 ***
(3) Casual	0.505 ***	0.419	0.282 *	0.936 ***	0.662 ***	0.654 ***	0.647 ***
**Preference**							
(1) Prefer to work same							
(2) Prefer to work less	0.277 ***	0.346 ***	0.324 **	0.428 *	0.211 *	0.037	0.361 *
(3) Prefer to work more	−0.215 *	−0.7 **	−0.678 ***	−0.779 *	0.97	−0.3	0.454 **
**Job dissatisfaction**							
(1) Unsatisfied on job payment	0.078	0.146	0.197	−0.076	0.026	−0.144	0.172
(2) Unsatisfied on job security	0.631 ***	0.305	0.29	0.342	0.917 ***	0.976 ***	0.858 ***
(3) Unsatisfied on work itself	0.449 ***	0.064	0.068	0.046	0.738 ***	0.68 ***	0.798 ***
(4) Unsatisfied on working hours	0.094	0.238	0.281 *	0.12	−0.007	0.283	−0.256
State average unemployment rate	0.063	0.069	0.045	0.145	0.048	0.102	0.001
Constant	−3.044	−3.773	−4.458	−4.863	−4.079	−5.372	−4.543
Observations	16811	16756	16756	16811	16756	16756	16811

Source: Authors’ own estimations using the pooled data of HILDA survey 2002–2011. Notes: (1) * significant at 10%; ** significant at 5%; *** significant at 1%. (2) The eligible age for superannuation is not controlled into the final regression model as it is highly related to age and age square term.

**Table 4 ijerph-16-03769-t004:** Summary of significant predictors of voluntary and involuntary work exits.

Predictors	Voluntary Work Exits	Involuntary Work Exits
Individual and household characteristics	Age (+), currently no partner (−), number of dependent students (−)	Insignificant
Health status	Long term health conditions (+)	Long term health conditions (+)
Financial concerns	Paying off mortgage (−), partner’s working (−) and partner’s income (+)	partner’s working (−)
Employment history	Tenure in current occupation (+), proportion of paid years (−),	Tenure in current occupation (−), proportion of paid years (−), proportion of unemployed years (+)
Work conditions	Part time (+), casual (+), prefer to work less (+), prefer to work more (−),	public sector (−), fixed term (+), casual work (+), prefer to work less (+),
Job dissatisfaction		dissatisfied with job security (+), dissatisfied with work itself (+)

Note: Summary of estimated signs of predictors from the regression Model 2 presented n Table 3. Data: HILDA survey 2002–2011.

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
