# Peer review of "Factors Predicting Voluntary and Involuntary Workforce Transitions at Mature Ages: Evidence from HILDA in Australia"

_ijerph, 2019, doi:10.3390/ijerph16193769_

Round 1
Reviewer 1 Report
The authors try to find relevant predictors for workforce transitions especially for older working people. They use a longitudinal survey for this purpose.
The manuscript is ambitious on one hand to find relevant answers on the other hand there are severe critical points in this text.
A first relevant aspect is that the authors do not fully explain their theoretical standpoint. At page three at the third paragraph they write about the approaches. This section is very short. It is also very doubtful whether the fit-variables really can be seen in this data set. The next approach of the “elderly migration” model also possibly cannot be seen in the variables of the data set. At least the authors have to explain whether there is a “fit” between the variables/concepts they explain and the variables which are the operationalized in the data set they use. The following “hypothesis” is in fact exploratory as there is no real hypothesis which can be proven.
The authors are possibly not involved in the longitudinal survey HILDA. Therefore, they are (possibly) not responsible for the data acquisition. Nevertheless, it would be interesting to know more about the background and also about the variables. The description is very scarce. As this is the central data acquisition there is at first no hint to a literature which can be found and even then, it should be included better.
Starting here it can be found that central references like Wooden and Watson (2007) are not included in the reference list. This is true for other references like Hatcher (2003) and Poehl et al. 2011). The references could be found but, sorry to write this, this does not look like a careful preparation of the manuscript.
The very important variable job satisfaction is especially not described well enough. As the authors are using it as a central variable this needs more information. It is not clear why job satisfaction is measured as one single item and later there are four different dimensions (payment, security, work, hours). It should be presented already in the section methods how this is measured. Possibly the survey study authors have reliability values of the measurement and could present them.
The usage of job satisfaction has to be explained more deeply as this is typically a variable which explains turnover to a high degree. The authors make an artificial split at the cutpoint of 5. There is no rationale for that. Later it can be read that 1/5 of the workers are dissatisfied. This leads to the assumption of a skewed variable. A simple median split possibly could have been more fruitful.
The predicting factors for workforce transitions look a bit arbitrary. The authors introduced the possible models but, as written, this is too short to understand this selection.
Especially the known fact of risk factors at work, i.e. physical aspects (hazardous work, impairing work) and psychosocial risk factors are not included. This should be discussed at least. Although subjectively sometimes the work ability index is used to assess the results of critical work.
The main critical point of this study is the usage of the pooled work transitions. As the authors describe first, the data stem out of a longitudinal survey. It is not presented how many participants stay in the long run or whether there are exchanges in the participating persons. Is the sum of the participants in every wave the sum of the lines in Table 1? They argue that the number of changes every year is low and sum up the persons for the nine waves. As this is a total confounding of persons which are changing and others which are not in the different waves this is methodological very critical and it should have been explained in detail why this should be possible and useful (in my point of view it is not).
First, there could be a confounding of persons who are changing very often. Then a sum of data which is dependent (longitudinal data!) is not acceptable. There are not 1182 people which are exiting their job as we can read that this are only approx. 120 persons per year. There can be an overlap and this can be the opposite.
Therefore, it is not understandable how the regression has been made and also under the assumption of using the data as independent (as in Table 2) the resulting analyses are doubtful.
As the authors do not write anything about the usage of the pooled data the further analyses are regarded as critical and the following results and discussion have not been viewed.
Possibly there is a methodological reason or way to pool the data but his has to be presented very clear. At the moment this is lacking.
Minor
The authors write about “the Australian literature review”. What is meant by that? Does this regard to the survey of HILDA? Other research results should be included too.
The description of the presentation of the manuscript at page three para six is redundant as this is a typical empirical approach of studies.
Is there really no mandatory retirement age in Australia? In other western countries in Europe this still exists.
In Figure 1 the y-axis is labelled job transition but this value is not explained in the text nor in the legend. The figure itself should be self-explaining.
Reviewer 2 Report
The topic of the paper is very interesting,
The introduction is well written.
My concerns are as follows:
1. Abstract: The part 3 (Results) is not comprehensive and the major findings should be presented.
2. Conclusion: It’s too long and the authors should make conclusion according to the results and discussion of this paper.
3. The paper has some errors, and could benefit from a proofreading. For example,
On page 5, line 8, “(136)” and “(274)” should be deleted;
The third paragraph below the “3.2. Individual characteristics and work conditions”, line 4, ”1 32 percent” should be corrected, line 5, the number “2” should be deleted.
The first paragraph below the “Summary of findings”, line 8, the number “4” should be deleted.
Reviewer 3 Report
This study analyzes the factors that determine the choice to continue working (or not) in older workers, and influence voluntary and involuntary work transitions. The different policies for older workers are compared and the authors draw from the comparison interesting observations.
The work is clear and well written and could only have minor improvements. It may be useful to mention the role of health promotion programs. It would be interesting to know if the obstacles and future prospects that health promotion programs for older workers encounter in Europe are also present in Australia (see: Magnavita N. Obstacles and Future Prospects: Considerations on Health Promotion Activities for Older Workers in Europe. Int J Environ Res Public Health. 2018 May 28;15(6). pii: E1096. doi: 10.3390/ijerph15061096).
Author Response
This study analyzes the factors that determine the choice to continue working (or not) in older workers, and influence voluntary and involuntary work transitions. The different policies for older workers are compared and the authors draw from the comparison interesting observations.
The work is clear and well written and could only have minor improvements. It may be useful to mention the role of health promotion programs. It would be interesting to know if the obstacles and future prospects that health promotion programs for older workers encounter in Europe are also present in Australia (see: Magnavita N. Obstacles and Future Prospects: Considerations on Health Promotion Activities for Older Workers in Europe. Int J Environ Res Public Health. 2018 May 28;15(6). pii: E1096. doi: 10.3390/ijerph15061096).
Thanks for sharing this reference with us. We have now added one paragraph as below to discuss this reference. The common obstacle of health promotion programs in Australia and Europe is how to ensure continuity of funding and effectiveness after the short term intervention.
“In Europe where some of countries have already taken important steps to tackle the challenge of ageing population, health promotion activities in workplace for ageing workers have recently been promoted as a new approach to improve occupational and populational health (Magnavita, 2018). Besides the historic approach that takes into account occupational risks, technical and medical expertise, and ergonomic adaptions in the work environment, this new approach promotes healthy habits which may delay the onset of diseases or help to manage the chronic diseases. A great effort has been made to increase the motivation of older workers to move to healthier habits, as older workers are more reluctant to make changes than younger workers. The existing health interventions in Australia have mainly occurred in communities for promoting physical activities and in primary care centers for disease prevention, as well as in health and aged care institutions for managing disabilities and chronic diseases, the health promotion activities in workplace are still inadequate. Australia shares the same obstacle for health promotion programs as in Europe on how to ensure continuity of funding and effectiveness after the end of the intervention programs.”
Round 2
Reviewer 1 Report
A first relevant aspect is that the authors do not fully explain their theoretical standpoint. At page three at the third paragraph they write about the approaches. This section is very short. It is also very doubtful whether the fit-variables really can be seen in this data set. The next approach of the “elderly migration” model also possibly cannot be seen in the variables of the data set. At least the authors have to explain whether there is a “fit” between the variables/concepts they explain and the variables which are the operationalized in the data set they use. The following “hypothesis” is in fact exploratory as there is no real hypothesis which can be proven.
The theory and conceptual approaches provide us theoretical framework to guide our choices of important predictors, and we are not aiming to test any of the theory and conceptual approaches.
We have now added more explanations on how these theoretical concepts can be linked to the variables selected from the HILDA survey data for our study. See the changes in “2.4 Selected predictors”.
There seems to be a strong different view of science. Either the work is exploratory or it is testing hypotheses. The whole study looks for predictors and uses the results in a way as if there were some hypotheses before. This critical point is still the case. The text did not change in any way in this point. So, this has to be considered and changed resp. adapted.
Again. If the study is done exploratory, this has to be declared. If it is theory driven, as it looks like with 2.4, then there have to be clear stated hypotheses which are related in the results part.
The authors are possibly not involved in the longitudinal survey HILDA. Therefore, they are (possibly) not responsible for the data acquisition. Nevertheless, it would be interesting to know more about the background and also about the variables. The description is very scarce. As this is the central data acquisition there is at first no hint to a literature which can be found and even then, it should be included better.
We have now provided more explanations on job satisfaction/dissatisfaction and long term health conditions. See the changes in “2.4 Selected predictors”.
This is the case. Therefore, there are new question, especially regarding JS, see later.
Starting here it can be found that central references like Wooden and Watson (2007) are not included in the reference list. This is true for other references like Hatcher (2003) and Poehl et al. 2011). The references could be found but, sorry to write this, this does not look like a careful preparation of the manuscript.
We have actually cited the three references in the main text: Wooden and Watson (2007), Hatcher (2003) and Poehl et al. 2011). However, we are sorry that we have made a mistake in rush on the list of references which is found to be for my another paper. We have now updated the right list of references for this paper.
We have now carefully checked the consistence between the citations in the main text and the list of references.
OK.
The very important variable job satisfaction is especially not described well enough. As the authors are using it as a central variable this needs more information. It is not clear why job satisfaction is measured as one single item and later there are four different dimensions (payment, security, work, hours). It should be presented already in the section methods how this is measured. Possibly the survey study authors have reliability values of the measurement and could present them.
The usage of job satisfaction has to be explained more deeply as this is typically a variable which explains turnover to a high degree. The authors make an artificial split at the cutpoint of 5. There is no rationale for that. Later it can be read that 1/5 of the workers are dissatisfied. This leads to the assumption of a skewed variable. A simple median split possibly could have been more fruitful.
We have explained more on our measure “job satisfaction” as below.
Job satisfaction in HILDA is measured by a group of questions asking individuals “How satisfied are you with your job (overall, job payment, job security, flexibility, work itself and working hours) on a scale of 0 (the most dissatisfied) to 10 (the most satisfied)?” In this study, we are interested to know the impact of job dissatisfaction with 4 job aspects (instead of the satisfaction level or overall job satisfaction) on workforce transitions. Consequently, we generate a dummy variable for job satisfaction: dissatisfied if with a response from 0 to 5 and satisfied if with a response from 6 to 10 as normally 60 out of 100 score is perceived as satisfied in school examines. We did not use “below median or mean” as it does not necessary indicate unsatisfied.
This can be done like that but there is in fact no explanation why this dividing has been used. There is no information about the distribution of the dimensions of JS.
The predicting factors for workforce transitions look a bit arbitrary. The authors introduced the possible models but, as written, this is too short to understand this selection.
We have now added more explanation on how we chosen the important predictors into our final models based on the theoretical framework and previous studies. See the changes in “2.4 Selected predictors”.
The chapter 2.4 is a presentation of the background ideas and therefore do not belong to the Methodology. Exactly this information would be expected in the introduction part which leads to clear hypotheses (see before).
Especially the known fact of risk factors at work, i.e. physical aspects (hazardous work, impairing work) and psychosocial risk factors are not included. This should be discussed at least. Although subjectively sometimes the work ability index is used to assess the results of critical work.
Thanks for the comments. We have now discussed these in the discussion section as below.
“Other limitations of this study include the utilization of 1-digit occupational group to identify whether the work is physically demanding, and the use of job dissatisfaction to identify potential psychosocial risk factors in the workplace. For future research, more detailed occupational categories in HILDA can be used to identify possible impacts of hazardous work and impairing work on work force transitions. In addition, work-related stress (Leka et al 2004) and workplace psychological harassment ((Grazia et al 2003) are recognized world-wide as major challenges to workers' health and the health of their organizations. Mental health problems and other stress-related disorders are known to be among the leading causes of early retirement from work, high absence rates, overall health impairment (Grazia et al 2003; Leka et al 2004). but larger cross-sectional and longitudinal survey for mature aged and older workers are necessary for further estimations.”
Ok.
The main critical point of this study is the usage of the pooled work transitions. As the authors describe first, the data stem out of a longitudinal survey. It is not presented how many participants stay in the long run or whether there are exchanges in the participating persons. Is the sum of the participants in every wave the sum of the lines in Table 1?
We have now added one note under Table 1:
“The first part of this table presents the number of work force transitions by staying at work, voluntarily or involuntarily not working occurred in every year; the second part of this table reports the number of voluntary or involuntary work force transitions by whether back to work until last wave (year 2011). “
As shown in the first part of Table 1, there are about 1100 total workforce transitions occurred in 10 years (or 106-161 transitions in every year). The sum of the participants in every wave is the sum of the lines in Table 1.
As shown in the second part of Table 1, we have also identified whether people had ever returned to paid work until the last recorded HILDA wave (Wave 11 in year 2011) after each of the work force transitions.
I still cannot understand your tables and this will be the case for the readers.. Where do I find the “1100 total wf transitions” in the table? Does “Lest paid work” mean “Left paid work”? Does “Involuntarily left paid work until last wave” belong to the “second part” of the table? It would be also much better to make two tables for reference purposes. Or is the “2nd part” a prolongation of the first? How to read this table?
They argue that the number of changes every year is low and sum up the persons for the nine waves. As this is a total confounding of persons which are changing and others which are not in the different waves this is methodological very critical and it should have been explained in detail why this should be possible and useful (in my point of view it is not).
First, there could be a confounding of persons who are changing very often. Then a sum of data which is dependent (longitudinal data!) is not acceptable. There are not 1182 people which are exiting their job as we can read that this are only approx. 120 persons per year. There can be an overlap and this can be the opposite.
Therefore, it is not understandable how the regression has been made and also under the assumption of using the data as independent (as in Table 2) the resulting analyses are doubtful.
As the authors do not write anything about the usage of the pooled data the further analyses are regarded as critical and the following results and discussion have not been viewed.
Possibly there is a methodological reason or way to pool the data but his has to be presented very clear. At the moment this is lacking.
We have now added more discussion on why using pooled data from longitudinal survey .
Our methodology recognizes that people at mature ages can leave the workforce for a time and then return to paid work. We count each transition from working in one year to not working in the subsequent year as an independent transition, while the change from one job to another is counted as no transition group “stayed at work”. For example, if an individual worked in one year, left work in the subsequent year, and returned to work before or until the last wave (either at a same or different job), we counted as one transition. If an individual worked in one year, left work, returned to work and left work again before or until the last wave, we counted as two transitions. Each individual can have multiple transitions over ten years and every transition is associated with changing age, family structure, financial situation and work conditions though gender and education might be stable over time.
The strategy of using pooled data drawn from different waves of existing longitudinal data could as below increase the number of transitions and hence be able to estimate the transition models that would not be possible by using any single year data with small sample size. As a large single cross-sectional study with workforce transitions for older workers aged 45 and over is not available in Australia, this approach is not only feasible, time- and cost-efficient but also permits the study of transitions occurred in a longer period than was assessed within any single investigation.
There might be concern on the autocorrelation and dependence of the residuals in our model generated by using the pooled data from longitudinal survey. We have adopted two ways to decrease this potential bias. Firstly, cross sectional weights in each year are applied to the pooled data in order to represent the national population. Secondly, we have controlled for as much as individual characteristics and took into account the reasons for leaving work and main activities after workforce transitions in defining voluntary and involuntary not working. Nonetheless, we estimate predictors and advise caution in any attempt to interpret our results in terms of causality (Gong and Kendig 2018). All the analyses are conducted by STATA 15.1.
The presented points are here for the reviewer, but not for readers. The only change I can see regards a short Note at the table 1 (which does not clarify the things). The critical point still remains and is also not explained in any way in the text itself: “First, there could be a confounding of persons who are changing very often. Then a sum of data which is dependent (longitudinal data!) is not acceptable. There are not 1182 people which are exiting their job as we can read that this are only approx. 120 persons per year. There can be an overlap and this can be the opposite.”
Minor
The authors write about “the Australian literature review”. What is meant by that? Does this regard to the survey of HILDA? Other research results should be included too.
“The Australian literature review….” Looked at previous studies in Australia only. While the paragraph before this provides “A brief international literature review…” including literature on both Australia and other countries.
Ok, partly. It still is not clear why there is a difference between an international and so-called national literature review. Both should be done at the same time.
In Figure 1 the y-axis is labelled job transition but this value is not explained in the text nor in the legend. The figure itself should be self-explaining.
We have now revised Figure 1 using “workforce transitions” to make it be self-explaining and consistent with the definitions used in the main text.
OK, this improved a lot.
Author Response
Thanks again for all the constructive comments. We have addressed your comments carefully and revised the main text accordingly as highlighted in blue colour.
